# Racial disparities in continuous glucose monitoring-based 60-min glucose predictions among people with type 1 diabetes

Helene Bei Thomsen[1,2], Livie Yumeng Li[1,2], Anders Aasted Isaksen[2],
Benjamin Lebiecka-Johansen[2], Charline Bour[3,4], Guy Fagherazzi[3],
William P. T. M. van Doorn[5,6], Tibor V. Varga[7], Adam Hulman[1,2]*

1 Department of Public Health, Aarhus University, Aarhus, Denmark, 2 Steno Diabetes Center Aarhus, Aarhus, Denmark, 3 Department of Population Health, Luxembourg Institute of Health, Strassen, Luxembourg, 4 Université Grenoble Alpes, Fonds de Dotation Clinatec, Grenoble, France, 5 CARIM School for Cardiovascular Diseases, Maastricht University, Maastricht, The Netherlands, 6 Department of Clinical Chemistry, Central Diagnostic Laboratory, Maastricht University Medical Centre+, Maastricht, The Netherlands, 7 Copenhagen Health Complexity Center, Department of Public Health, University of Copenhagen, Copenhagen, Denmark

* adahul@rm.dk

## Abstract

Non-Hispanic white (White) populations are overrepresented in medical studies. Potential healthcare disparities can happen when machine learning models, used in diabetes technologies, are trained on data from primarily White patients. We aimed to evaluate algorithmic fairness in glucose predictions. This study utilized continuous glucose monitoring (CGM) data from 101 White and 104 Black participants with type 1 diabetes collected by the JAEB Center for Health Research, US. Long short-term memory (LSTM) deep learning models were trained on 11 datasets of different proportions of White and Black participants and tailored to each individual using transfer learning to predict glucose 60 minutes ahead based on 60-minute windows. Root mean squared errors (RMSE) were calculated for each participant. Linear mixed-effect models were used to investigate the association between racial composition and RMSE while accounting for age, sex, and training data size. A median of 9 weeks (IQR: 7, 10) of CGM data was available per participant. The divergence in performance (RMSE slope by proportion) was not statistically significant for either group. However, the slope difference (from 0% White and 100% Black to 100% White and 0% Black) between groups was statistically significant (p = 0.02), meaning the RMSE increased 0.04 [0.01, 0.08] mmol/L more for Black participants compared to White participants when the proportion of White participants increased from 0 to 100% in the training data. This difference was attenuated in the transfer learned models (RMSE: 0.02 [-0.01, 0.05] mmol/L, p = 0.20). The racial composition of training data created a small statistically significant difference in the performance of the models, which was not present after using transfer learning. This demonstrates

**Data availability statement:** The dataset was a public available dataset retrieved from https://public.jaeb.org/dataset/542 (RacialDifferences_2024-02-22.zip, T1DX Racial Glycation Gap Protocol 09-04-14 Final-V1.3). The analysis content and conclusions presented in the study are solely the authors' responsibility and have not been reviewed or approved by the T1D Exchange Clinic Network site, Helmsley Charitable Trust, or the 'Racial Differences in Mean CGM Glucose in Relation to HbA1c' project. Python code for data processing, model development and R code for the linear mixed effect models can be found on GitHub at: https://github.com/hulmanlab/jchr_racial_diff.

**Funding:** HBT, LYL, AAI, BLJ, and AH are supported by a Data Science Emerging Investigator grant (NNF22OC0076725) by the Novo Nordisk Foundation. Furthermore, this work was supported by a research grant from the Danish Diabetes and Endocrine Academy and the Danish Cardiovascular Academy, which are funded by the Novo Nordisk Foundation, grant numbers NNF22SA0079901 and NNF20SA00657242 through HBT's PhD scholarship. TVV is supported by the "Data Science Investigator - Emerging 2022" grant from the Novo Nordisk Foundation (NNF22OC0075284). CB is supported by OCIRP and AGRICA. The funders had no role in study design, data collection and analysis, decision to publish, or preparation of the manuscript.

**Competing interests:** The authors have declared that no competing interests exist.

the importance of diversity in datasets and the potential value of transfer learning for developing more fair prediction models.

## Author summary

Non-Hispanic White populations are often overrepresented in medical datasets. Training machine learning models on such data may lead to unfair clinical prediction tools and an unfavorable impact on healthcare inequalities. This study investigated how well machine learning models perform in predicting blood sugar levels for Non-Hispanic White and Non-Hispanic Black people with type 1 diabetes. We used continuous glucose monitoring (CGM) data from people with type 1 diabetes living in the US to compare various methods and models trained on datasets with different proportions of White and Black participants. We found a difference between the performance improvement in White and the performance drop in Black participants as the proportion of White participants increased in the dataset used for training. This difference disappeared when models were further tailored to individuals. Our work demonstrates the importance of using diverse training data when developing AI-based solutions for healthcare.

## Introduction

Type 1 diabetes is an autoimmune chronic metabolic disease characterized by chronic high blood glucose levels [1,2]. Keeping blood glucose levels within normal range is essential to avoid future diabetes-related complications, such as cardiovascular diseases, blindness, renal failure, and lower extremity amputations [2–5].

Automated insulin delivery systems have been developed for blood glucose management [6]. These systems generally utilize blood glucose prediction models to forecast future blood glucose values using continuous glucose monitoring (CGM) devices [6]. Several algorithms exist to calculate or adjust insulin infusion based on CGM data. The simplest is low-suspend glucose systems, which automatically suspend insulin infusions when blood glucose levels fall under a given threshold [7]. Other more advanced methods calculate the insulin infusion rate as a function of time and consider other factors, such as carbohydrate intake, physical activity, and stress [6,8]. Machine learning models have become increasingly popular because of the increased data availability [6]. The latest advancements within this field are based on artificial neural networks, which do not require manual extraction of features from time series data [6]. However, machine learning models might propagate biases observed in the training data [9]. The majority of US trials do not report on race (57%), and despite a slight improvement over time, racial minority groups are underrepresented in trials (20%) [10]. Moreover, Muralidharan et al. found that less than 4% of the FDA summary documents provide any information about race/ethnicity [11]. This lack of documentation on race/ethnicity poses challenges since studies

have found that physiological differences such as insulin resistance, insulin sensitivity, glycaemic response, and glycaemic control vary by ethnicity [12–14]. Furthermore, racial/ethnic minorities with type 1 diabetes have a higher burden of diabetes-related complications in the US [5]. These disparities across racial and ethnic groups, coupled with the underrepresentation of minority groups in datasets, can lead to biased prediction models favoring the majority population. Cronjé et al. found that risk prediction algorithms for type 2 diabetes often underestimate the risk for non-Hispanic Black individuals [15]. A study by Obermeyer et al. found that closing this gap in prediction models significantly increased the resources allocated to Black patients, from 18% to 47% [16]. Even though assisting technologies, such as clinical prediction models, are expected to provide equal access to care, they can lead to increasing health inequality if not developed and implemented with care.

A practical approach to addressing data-based healthcare disparities and reducing bias in prediction models is to fine-tune an existing or pre-trained model to specific populations, utilizing transfer learning techniques [9,17]. Fine-tuning allows models trained on general datasets to adapt to the unique characteristics of underrepresented groups, improving accuracy and reducing bias by leveraging existing knowledge embedded in models. This approach has also been used to predict blood glucose levels [18–20]. These studies explored fine-tuning to personalize models with different prediction horizons and across different types of diabetes. However, none of them considered algorithmic fairness and racial disparities.

Our study aims to evaluate the fairness of algorithms in glucose prediction and compares different strategies for tailoring prediction models to individuals.

## Methods

### Study population and data collection

The data was collected by the T1D Exchange non-profit organization from diabetes centers in the United States [21]. The original study included 208 people with T1D [13]. We excluded three participants who did not complete the study, resulting in 205 participants included. Among them, 101 were non-Hispanic White (White), and the other 104 were non-Hispanic Black (Black). Race and ethnicity were defined as self-identified and self-reported using a race/ethnicity verification form [13]. The ethnicity category defined Hispanic or Latino as "a person of Cuban, Mexican, Puerto Rican, South or Central American, or other Spanish culture or origin, regardless of race. The term "Spanish origin" can be used in addition to "Hispanic" or "Latino"" [13]. To identify the race of the participants, the study [13] defined White as "a person having origins in any of the original peoples of Europe, the Middle East or North Africa." and Black or African American as "a person having origins in any of the Black racial groups of Africa. Terms such as "Haitian" or "Negro" can be used in addition to "Black" or "African American"" [13]. Furthermore, both parents had to be non-Hispanic White, or both parents had to be non-Hispanic Black if parental ethnicity and race were known. In our study, the term race will be used moving forward, as it aligns with the terminology used by Bergenstal et al. [13]. However, the authors acknowledge/appreciate that race is a complex and challenging construct to define [22]. Two age groups were sampled for each race: 8 to under 18 years old and 18 years and older. Each participant wore a blinded, modified investigational version of Abbott Diabetes Care's Flash Glucose Monitoring System (CGM) for 12 weeks. CGM data was sampled every 15 minutes within a range of 2.21 and 27.65 mmol/L (40 and 498 mg/dL) [13]. Measurements exceeding those bounds were recorded as 2.21 or 27.65 mmol/L (40 and 498 mg/dL). The study adhered to the tenets of the Declaration of Helsinki and was approved by the local institutional review boards. Study participants provided written informed consent and assent when applicable [13].

### Study design

Data was grouped in 60-minute observational windows, including four measurements ($x_1$, $x_2$, $x_3$, $x_4$) to predict the blood glucose value 60 minutes ahead ($x_8$) (Fig 1). For each participant, eleven training sets were created, each including 100 other participants' data with varying proportions of White and Black participants (0–100%, 10–90%,..., 100-0%). This

PLOS Digital Health

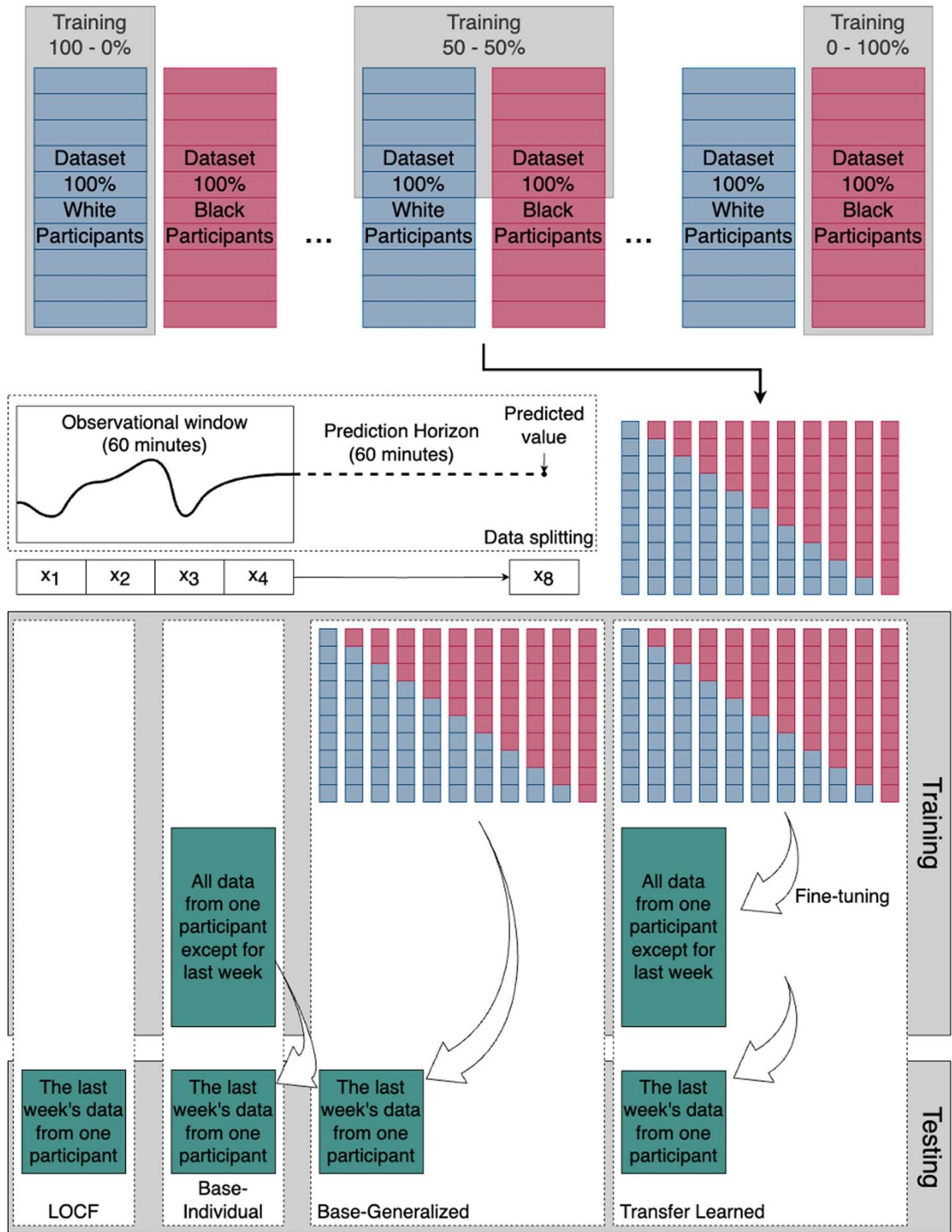

**Fig 1. Workflow for evaluating prediction models using stratified datasets of non-Hispanic White and non-Hispanic Black participants.** The top panel illustrates progressive training strategies with varying proportions of White and Black patients (from 100%-0% to 0%-100%) marked by the

grey rectangles behind the datasets. Samples are created from 60-minute observational windows and prediction horizons (60 minutes). The lower panel outlines four modeling approaches: Last observation carried forward (LOCF), Base-Individual (training on all data except the last week from a single participant), Base-Generalized (training on data from multiple participants), and Transfer Learning (fine-tuning a generalized model on individual data).

resulted in 2,255 training datasets (205 individuals times 11 training sets with varying proportions of race) and 205 independent test sets. By creating 11 subsets of training data for each participant in the dataset, we ensured there was no data leakage from each participant to the 11 training datasets assigned to them. This is important since data from each participant was used for transfer learning, and the last week's worth of CGM data was used for testing.

Four training approaches were used when developing the prediction models for each participant.

1. Last observation carried forward (LOCF) - naive baseline: In the LOCF model, a given CGM measurement ($x_4$) was used to predict blood glucose values 60 minutes in advance ($x_8$) in the test set by assuming no change in the last 60 minutes.

2. Base-Individual: In the Base-Individual model, each participant's complete CGM data (except for the last week, which was used for the testing data) was used for training.

3. Base-Generalized: Here, for each participant, the 11 training corresponding datasets described above were used to generate 11 separate models, which were all subsequently tested on the given participant's last week of CGM data.

4. Transfer Learned: This model mirrored the Base-Generalized Model's setup. However, the 11 trained models for each participant were subsequently tailored, i.e., fine-tuned with transfer learning to the given participant's complete CGM data (except for the last week, which was used for the testing data) (Fig 1).

## Model selection and training

A long short-term memory model (LSTM) was built using the Keras library version 2.15.0 with Python version 3.11.4. The overall architecture (two LSTM layers followed by a dense layer) was previously described in van Doorn et al. [23]. Glucose values for the training input were scaled to be within the range of zero to one with the MinMaxScaler function from scikit-learn library version 1.3.0. Hyperparameters included the number of neurons in LSTM layers, activation functions, dropout rate, batch size, learning rate, decay rate, and early stopping. These hyperparameters were tuned using a grid search, heatmaps, and manual tuning. Further information is given in the code repository.

## Model evaluation

The models were evaluated internally using each participant's last week's worth of data, corresponding to 672 measurements from each participant. The root mean squared error (RMSE) was calculated for each participant and proportion as a metric of predictive performance, which resulted in 101 RMSEs for White and 104 RMSEs for Black participants for each proportion. The mean (95% confidence interval [CI]) was calculated for both groups. This was done for all four prediction model approaches. A linear mixed effect model was used to investigate whether there were any changes in the predictive performance based on the training dataset with varying proportions of White and Black participants, on which the Base-Generalized and Transfer Learned models were trained. The linear mixed effect model approach was chosen because data points from the same individual cannot be considered independent measurements. A random intercept was included for individuals since it was assumed there would be variation in RMSE results between participants. The variance of the random intercepts describes between-person variability. The model included the following covariates: ratio (proportion of White and Black), race, age, sex, and training data sample size. An interaction term between race and ratio was added to model a potential divergence in performance by race because it was assumed training data composition affected

performance. This means that a different slope was allowed for the two racial groups when the linear mixed effect models were fitted to the results so the hypothesis could be tested. Race, age, and sex were categorical variables, and RMSE, ratio, and sample size were continuous variables. The linear mixed effect mode was built using R version 4.4.1, the lme4 package version 1.1.35.4, and the Epi package version 2.51.

A surveillance error grid was used to evaluate the prediction models' accuracy according to risk zones [24,25]. The methcomp package version 1.0.0 was used to calculate the risk zones in this study [23]. Each CGM prediction error for the test set was categorized as: "No", "Slight", "Moderate", "High", and "Extreme" errors according to the absolute error between the predicted and the actual value and the risk zone the error occurred in (Fig 2). Predictions outside of 0–33 mmol/L were changed to 0 or 33 mmol/L (0 or 594 mg/dL) to keep within the upper and lower bounds of the surveillance error grid. The zones were calculated for each ratio (0–100%, 10–90%,..., 100-0%) for each model.

## Results

Baseline characteristics are summarized in Table 1. A median of 9.5 [IQR: 7.7, 10.7] weeks of CGM data was available for White participants and 8.6 [IQR: 6.9, 9.9] weeks for Black participants. There were 49 children out of the 101 White participants and 33 out of the 104 Black participants. There was a higher proportion of females in both groups.

The predictive performance of the models is shown in Table 2 and Fig 2. Naive (LOCF) and Base-Individual models performed the worst. Errors were about 0.5 mmol/L lower for the Base-Generalized and Transfer Learned models. For white participants, the Base-Generalized model trained on a dataset including exclusively Black participants had an RMSE of 2.04 [1.95, 2.13], while its performance was 2.01 [1.92, 2.10] when trained exclusively on White participants (difference: -0.003 [-0.007, 0.001], p=0.12, Table 3). For the same two proportions, the Transfer Learned models showed RMSEs of 2.02 [1.94, 2.11] and 2.01 [1.93, 2.10] mmol/L, respectively (difference: -0.001 [-0.003, 0.001], p=0.46, Table 4). For the Base-Generalized models, there was a statistically significant interaction between the effect of race and ratio as the RMSE increased 0.04 [0.01, 0.08] mmol/L more for Black participants compared to White participants when the proportion of White participants increased from 0 to 100% in the training data (Table 3). We did not find evidence for this divergence in performance in the Transfer Learned models (p=0.20, Table 4). Furthermore, statistically significant differences were found between age groups in both the Base-Generalized (0.482 [0.355, 0.610], p<0.001, Table 3) and the Transfer Learned models (0.411 [0.290, 0.533], p<0.001, Table 4) showing higher error when predicting on children compared to adults. For every 1,000 increase in sample size for fine-tuning the Base-Generalized model to an individual (transfer learning), the RMSE decreases with -0.111 [-0.161, -0.061] mmol/L (p<0.001, Table 4). At the same time, we did not find an

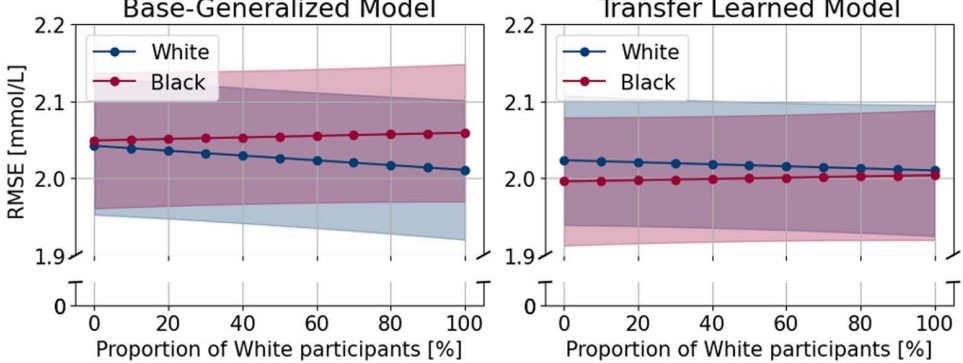

**Fig 2. Estimated root mean squared errors (RMSE) across 11 different proportions of White and Black participants in the dataset.** Note: the y-axis starts at 1.9 mmol/L to visually show the difference between the red and the blue lines better.

**Table 1. Baseline characteristics table.**

| Characteristics | Race | | |
|---|---|---|---|
| | Total (N = 205) | White (N = 101) | Black (N = 104) |
| Race (White) | 101 (49%) | 101 (100%) | 0 (0%) |
| Sex (male) | 88 (43%) | 46 (46%) | 42 (40%) |
| Age group (child, < 18 yrs) | 82 (40%) | 49 (49%) | 33 (32%) |
| Age (years) | 28.0 [15.0;47.0] | 22 [13, 46] | 32.0 [16.0, 47.2] |
| BMI (kg/m²) | 25.1 [21.7, 29.2] | 24.0 [20.7, 28.6] | 26.7 [22.3, 30.0] |
| Age at T1D diagnosis | 11.0 [7.0, 22.0] | 9.0 [6.0, 16.0] | 13.5 [8.0, 24.5] |
| HbA1c (mmol/mol) | 67 [57, 78] | 63 [55, 73] | 72 [60, 85] |
| CGM duration w/o missing data (weeks/participant) | 9.1 [7.2, 10.3] | 9.5 [7.7, 10.7] | 8.6 [6.9, 9.9] |
| Missing CGM values (weeks/per participant) | 3.3 [1.6, 5.1] | 2.6 [1.3, 4.5] | 4.1 [1.8, 5.6] |
| Time in range (%/per participant) | 47 [35, 57] | 49 [39, 60] | 42 [31, 54] |
| Time above range (%/per participant) | 45 [30, 56] | 42 [28, 53] | 48 [32, 60] |
| Time below range (%/per participant) | 8 [4, 14] | 6 [4, 10] | 11 [5, 15] |

Categorical values are given by number (percentage), and numerical values are given by median [interquartile range].

association between the sample size used for training the Base-Generalized model and RMSE (p = 0.62). Between-person variance of RMSE was 0.198 for the Base-Generalized and 0.172 for the Transfer Learned model.

Predictions from all models for all participants were evaluated with a surveillance error grid [24], which is illustrated in Fig 3 for one randomly chosen participant at one ratio and summarized in S1 and S2 Tables.

The results from the surveillance error grid showed that the naive approach, when assuming no change within the last 60 minutes, had 69% of the predictions in the no-risk area and 26% in the slight-risk area. The generalized models had 72–75% of the predictions in the no risk, and 22–25% were in the slight risk area of the grid. For the Transfer Learned models, 72% to 75% of the predictions were in no risk, and 22–25% were in the slight risk zones. All values for the none and slight risk zones only change with a maximum of one percent-point across the 11 proportions for both White and Black groups (S1 and S2 Tables).

## Discussion

### Main findings

Our study investigated how the racial composition of data from non-Hispanic White and non-Hispanic Black participants impacts the performance of machine learning models when predicting blood glucose levels in people with type 1 diabetes. Additionally, the study evaluated whether transfer learning can minimize performance disparities between racial groups. We found that the racial composition of training data led to a small statistically significant divergence in performance between the slopes that were fitted to the results from White and Black participants when models were not personalized. This suggests increasing disparity, with performance differences favoring White individuals becoming more pronounced as their proportion in the dataset grows. Although the difference between the performances of the two racial groups might not be large enough to produce clinical inequalities in this study, the findings still indicate that the machine learning models could pick up on racial markers within the CGM data that was used in our study. This stresses the importance of training data composition from the perspective of algorithmic fairness to avoid increasing disparities between ethnic, racial, and

**Table 2. Root mean squared errors with 95% CIs for the different models.**

| Model performance (RMSE) for White participants | | | | |
| --- | --- | --- | --- | --- |
| Ratio | LOCF | Base-Individual | Base-Generalized | Transfer Learned |
| 0 | | | 2.04 [1.95, 2.13] | 2.02 [1.94, 2.11] |
| 10 | | | 2.04 [1.95, 2.13] | 2.02 [1.94, 2.11] |
| 20 | | | 2.04 [1.95, 2.12] | 2.02 [1.94, 2.10] |
| 30 | | | 2.03 [1.94, 2.12] | 2.02 [1.94, 2.10] |
| 40 | 2.45 [2.32, 2.57] | 2.54 [2.38, 2.70] | 2.03 [1.94, 2.12] | 2.02 [1.94, 2.10] |
| 50 | (independent | (independent | 2.03 [1.94, 2.11] | 2.02 [1.93, 2.10] |
| 60 | of ratio) | of ratio) | 2.02 [1.94, 2.11] | 2.02 [1.93, 2.10] |
| 70 | | | 2.02 [1.93, 2.11] | 2.01 [1.93, 2.10] |
| 80 | | | 2.02 [1.93, 2.11] | 2.01 [1.93, 2.10] |
| 90 | | | 2.01 [1.92, 2.10] | 2.01 [1.93, 2.10] |
| 100 | | | 2.01 [1.92, 2.10] | 2.01 [1.93, 2.10] |
| **Model performance (RMSE) for Black participants** | | | | |
| Ratio | LOCF | Base-Individual | Base-Generalized | Transfer Learned |
| 0 | | | 2.05 [1.96, 2.14] | 2.00 [1.91, 2.08] |
| 10 | | | 2.05 [1.96, 2.14] | 2.00 [1.91, 2.08] |
| 20 | | | 2.05 [1.96, 2.14] | 2.00 [1.92, 2.08] |
| 30 | | | 2.05 [1.97, 2.14] | 2.00 [1.92, 2.08] |
| 40 | 2.41 [2.28, 2.53] | 2.63 [2.47, 2.79] | 2.05 [1.97, 2.14] | 2.00 [1.92, 2.08] |
| 50 | (independent | (independent | 2.05 [1.97, 2.14] | 2.00 [1.92, 2.08] |
| 60 | of ratio) | of ratio) | 2.06 [1.97, 2.14] | 2.00 [1.92, 2.08] |
| 70 | | | 2.06 [1.97, 2.14] | 2.00 [1.92, 2.08] |
| 80 | | | 2.06 [1.97, 2.14] | 2.00 [1.92, 2.09] |
| 90 | | | 2.06 [1.97, 2.15] | 2.00 [1.92, 2.09] |
| 100 | | | 2.06 [1.97, 2.15] | 2.00 [1.92, 2.09] |

Values that vary by proportion are estimated using linear mixed-effects models. These models were adjusted to the population's age (40% children) and sex (57% women) distribution. Abbreviations: LOCF = last observation carried forward.

demographic groups when considering integrating and using AI-driven healthcare tools. Performance differences disappeared when transfer learning was applied, demonstrating that the imbalance in training data can be mitigated with data from the underrepresented group in the initial training data.

Our study is one of the first to investigate racial disparities in blood glucose prediction models for type 1 diabetes and propose a method to address them through transfer learning. This is particularly relevant given that racial differences in insulin resistance, insulin sensitivity, and glycemic control have been documented [12]. Moreover, as diabetes technology is becoming more widely used [26], machine learning-driven glucose monitoring devices have already been FDA-approved [27,28], and racial disparities in prediction models have been found in the field of diabetes [15]. Further challenges that might increase racial disparities include unequal access to diabetes technology [29,30]. A study from the US found that racial disparities in CGM and insulin pump use among young adults with type 1 diabetes could not solely be explained by socioeconomic status, demographics, healthcare factors, and diabetes self-management [30].

Previous research found transfer learning to reduce healthcare disparities in a multi-ethnic cancer omics dataset, suggesting that transfer learning might be a potential solution to reduce healthcare disparities arising from data inequalities [9]. In our study, the Transfer Learned models outperformed the other approaches, which aligns with the current literature

**Table 3. Model coefficients from linear mixed effects models for root mean squared errors of the Base-Generalized models.**

| Predictor | Coefficient | 95% CI | P-value |
|---|---|---|---|
| Intercept | 1.893 | [1.762, 2.024] | <0.001 |
| Race | | | |
| White | Reference | | |
| Black | 0.007 | [-0.118, 0.132] | 0.91 |
| Age | | | |
| Adult | Reference | | |
| Child | 0.482 | [0.355, 0.610] | <0.001 |
| Sex | | | |
| Male | Reference | | |
| Female | -0.076 | [-0.201, 0.048] | 0.23 |
| Training size Base-Generalized (per 100,000 samples)* | 0.019 | [-0.055, 0.093] | 0.62 |
| Ratio (per 10%) | -0.003 | [-0.007, 0.001] | 0.12 |
| Ratio· Race | 0.004 | [0.001, 0.008] | 0.02 |

*Training size was centered at 510,000 as the training data had a mean of 514,293 samples.

[19,20]. Studies have found transfer learning to be superior to other methods when tailoring blood glucose predictions to different types of diabetes, testing different prediction horizons, and improving model performance for individuals with scarce data [19,20].

Mohebbi et al. found that a generalized model performed better than a transfer learned model; however, the difference in performance was 0.01 mmol/L in mean RMSE between the two models [31]. The authors speculated that the reason might have been the size of the datasets the models were fine-tuned on. This aligns with our findings, that the training sample size had a measurable impact on the performance of the Transfer Learned models by -0.1 mmol/L in RMSE

**Table 4. Model coefficients from linear mixed effects models for root mean squared errors of the Transfer Learned models.**

| Predictor | Coefficient | 95% CI | P-value |
|---|---|---|---|
| Intercept | 1.890 | [1.768, 2.015] | <0.001 |
| Race | | | |
| White | Reference | | |
| Black | -0.028 | [-0.147, 0.092] | 0.65 |
| Age | | | |
| Adult | Reference | | |
| Child | 0.411 | [0.290, 0.533] | <0.001 |
| Sex | | | |
| Male | Reference | | |
| Female | -0.057 | [-0.173, 0.060] | 0.34 |
| Training size Base- Generalized (per 100,000 samples)* | 0.011 | [-0.060, 0.081] | 0.77 |
| Training size tl (per 1,000 samples)** | -0.111 | [-0.161, -0.061] | <0.001 |
| Ratio (per 10 units) | -0.001 | [-0.003, 0.001] | 0.46 |
| Ratio· Race | 0.002 | [-0.001, 0.005] | 0.20 |

*Training size was centered at 510,000 as the training data had a mean of 514,293 samples.

**Training size for the transfer learned approach was centered at 4,400 as the training data for fine-tuning had a mean of 4,371 samples.

PLOS Digital Health

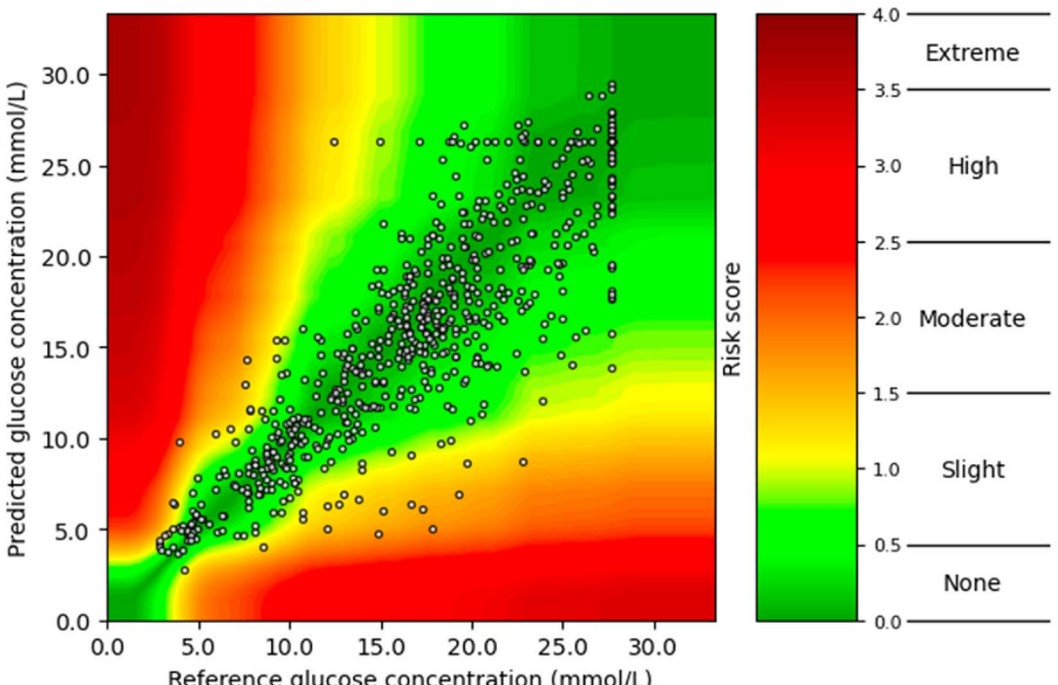

**Fig 3. Surveillance Error Grid of a Black, male, child participant for the Base-Generalized model trained exclusively on data from Black participants.**

per 1000 samples but not the Base-Generalized models. These findings highlight the importance of data availability on model performance. It is equally important to consider the clinical relevance of these performance differences. When using RMSE as a predictive performance metric, it is crucial to keep in mind that the severity of the predictions varies depending on where they are on the blood glucose scale. Half a mmol/L difference is not substantial if the models predict levels within the normoglycemic and hyperglycemic range; however, half a mmol/L difference can have immediate and severe consequences if the models predict levels in the hypoglycemic range. Multiple error grids have been developed to evaluate the performance of blood glucose monitors, such as self-monitoring with fingerprick [25]. In our study, the minimal changes in the percentages of the "none" and "slight risk" zones, along with the unchanged "moderate," "high," and "extreme risk" zones in the surveillance error grid, were observed between Base-Generalized and Transfer Learned models. These findings indicate that the differences in predictive performance between the models are unlikely to be clinically relevant. These results are similar to what van Doorn et al. reported in a similar study setup in people with type 1 diabetes [23]. They found 70% of the predictions in no risk and 28% in the slight risk area [23]. This highlights the complexity inherent in blood glucose prediction and the clinical relevance of error assessment beyond RMSE, especially with the increasing number of FDA approvals for AI/ML-enabled medical devices [32].

## Strength and limitations

The strength of this study is the innovative study design, which compares different strategies across different proportions of racial compositions in training data. The design can be reused for other data modalities beyond CGM data when evaluating fairness and composition of training data in prediction studies. Moreover, using an open-access dataset and sharing code strengthens the impact of the study, makes the findings reproducible, and provides valuable resources for future studies.

A limitation of this study is the size of the dataset, which influenced the statistical power of the analyses. The dataset only included non-Hispanic White and Black participants; however, there are other ethnic and racial minorities where racial disparities could be investigated. Other groups, such as age strata, could also be relevant to investigate in the future as our study found that predicting blood glucose levels for children is more challenging than for adults. Another limitation of our study is that data was limited to the United States and might not generalize well to other countries as diabetes management and care practices vary globally. Furthermore, having another dataset for external evaluation and a larger sample size could have further strengthened the results. Lastly, data was collected between 2014 and 2017, and there have been improvements in CGM devices since, which could have improved data quality.

## Conclusion and perspectives

The racial composition of the training data had a differential impact on the performance of glucose predictions by race. However, a transfer learning approach mitigated this issue. Our study highlights the importance of considering diversity in training data. Future studies on investigating algorithmic fairness by other sensitive attributes are warranted. Our study is a valuable resource for developing more fair diabetes technology in the future, which is essential now that diabetes technology is getting more widely used.

## Supporting information

**S1 Table. Results from Surveillance Error Grid for White participants.** Values in the table are given as absolute values (%).
(PDF)

**S2 Table. Results from Surveillance Error Grid for Black participants.** Values in the table are given as absolute values (%).
(PDF)

## Author contributions

**Conceptualization:** Helene Bei Thomsen, Anders Aasted Isaksen, Benjamin Lebiecka-Johansen, Adam Hulman.

**Formal analysis:** Helene Bei Thomsen, Livie Yumeng Li, Benjamin Lebiecka-Johansen, Charline Bour, Adam Hulman.

**Funding acquisition:** Helene Bei Thomsen, Adam Hulman.

**Methodology:** Helene Bei Thomsen, Livie Yumeng Li, Adam Hulman.

**Project administration:** Helene Bei Thomsen, Adam Hulman.

**Resources:** Livie Yumeng Li, Anders Aasted Isaksen, Charline Bour, William P. T. M. van Doorn, Tibor V. Varga, Adam Hulman.

**Supervision:** Anders Aasted Isaksen, Benjamin Lebiecka-Johansen, Guy Fagherazzi, Tibor V. Varga, Adam Hulman.

**Visualization:** Helene Bei Thomsen, Tibor V. Varga, Adam Hulman.

**Writing – original draft:** Helene Bei Thomsen.

**Writing – review & editing:** Livie Yumeng Li, Anders Aasted Isaksen, Benjamin Lebiecka-Johansen, Charline Bour, Guy Fagherazzi, William P. T. M. van Doorn, Tibor V. Varga, Adam Hulman.

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
