## [Decision Letter · Decision Letter 0]

21 Feb 2025

PDIG-D-25-00023Racial disparities in continuous glucose monitoring-based 60-min glucose predictions among people with type 1 diabetesPLOS Digital Health Dear Dr. Hulman, Thank you for submitting your manuscript to PLOS Digital Health. After careful consideration, we feel that it has merit but does not fully meet PLOS Digital Health's publication criteria as it currently stands. Therefore, we invite you to submit a revised version of the manuscript that addresses the points raised during the review process. Please submit your revised manuscript within 60 days Apr 22 2025 11:59PM. If you will need more time than this to complete your revisions, please reply to this message or contact the journal office at digitalhealth@plos.org. Please include the following items when submitting your revised manuscript:* A rebuttal letter that responds to each point raised by the editor and reviewer(s). You should upload this letter as a separate file labeled 'Response to Reviewers '. This file does not need to include responses to any formatting updates and technical items listed in the 'Journal Requirements' section below.* A marked-up copy of your manuscript that highlights changes made to the original version. You should upload this as a separate file labeled 'Revised Manuscript with Track Changes '.* An unmarked version of your revised paper without tracked changes. You should upload this as a separate file labeled 'Manuscript '. If you would like to make changes to your financial disclosure, competing interests statement, or data availability statement, please make these updates within the submission form at the time of resubmission. Guidelines for resubmitting your figure files are available below the reviewer comments at the end of this letter. We look forward to receiving your revised manuscript. Kind regards, Gloria Hyunjung KwakSection EditorPLOS Digital Health Gloria Hyunjung KwakSection EditorPLOS Digital Health Leo Anthony CeliEditor-in-ChiefPLOS Digital Healthorcid.org/0000-0001-6712-6626 **Journal Requirements:** **Additional Editor Comments (if provided):****Reviewers' Comments:** Reviewer's Responses to Questions

**Comments to the Author**

1. Does this manuscript meet PLOS Digital Health’s publication criteria ? Is the manuscript technically sound, and do the data support the conclusions? The manuscript must describe methodologically and ethically rigorous research with conclusions that are appropriately drawn based on the data presented.

Reviewer #1: Partly

Reviewer #2: Partly

2. Has the statistical analysis been performed appropriately and rigorously?

Reviewer #1: I don't know

Reviewer #2: Yes

3. Have the authors made all data underlying the findings in their manuscript fully available (please refer to the Data Availability Statement at the start of the manuscript PDF file)?

Reviewer #1: Yes

Reviewer #2: Yes

4. Is the manuscript presented in an intelligible fashion and written in standard English?

Reviewer #1: Yes

Reviewer #2: Yes

5. Review Comments to the Author

Reviewer #1: The authors study racial disparities in predicting blood glucose levels across time and the influence of the training dataset composition with respect to race on the error of the model. They find that one of their models shows slight evidence for a disparity between White and Black participants, while a transfer learned model does not.

While interesting, the main result of the study to me seems to be that race does not seem to be a major influence on the error of a blood glucose prediction model. Would the authors agree with that statement? Now, this could be because of the relatively small sample size or the comparably poor performance of all the models, or other factors, but this finding should be clearly stated and potential caveats discussed.

Regarding the statistical conclusions regarding the difference in slopes between White and Black participants, was this analysis determined before the data analysis started? Trying out different models may inflate the p-values, and I did not quite get the rationale behind the interaction described in line 169.

It seems the authors mainly do within subject model development. While this is interesting, it would be good to see how well the model transfers across participants. Is that possible at all? Is there anything that can be learned from one participant to the next?

Is 30 min really the relevant time scale? What about more finely binned data or data at the full native resolution of the measurement device?

Could the authors normalize each RMSE of the transfer or extended model by the respective performance of the base model in exactely that condition? Potentially this could reveal different effects now covered by the intersubject variability.

Minor points:

Line 94: “…study, 101 of whom were…” -> make new sentence, the connection with “whom” implies that 101 is a subset of 3.

Line 117: What is meant by “Eleven training sets were created for each participant in the dataset with varying proportions”. How can datasets created within one participant contain varying proportions of black and white participants? Clarify the sentence. Likewise, the formulation in line 122 is strange regarding the “11 training datasets assigned to them”. I would rather say that participants get assigned to training sets.

Line 243: Provide a citation for the surveillance error grid.

Reviewer #2: Brief description:

This study investigated algorithmic fairness in glucose prediction models using continuous glucose monitoring (CGM) data from 101 non-Hispanic White and 104 Black participants with type 1 diabetes. The research employs long short-term memory (LSTM) models trained with varying racial compositions in the training data and incorporates transfer learning techniques to predict glucose levels. The findings reveal an increase in root mean square error (RMSE) for Black participants as the proportion of White participants in the training data increases, highlighting the influence of racial composition on model performance. Additionally, this disparity diminishes when transfer learning is applied, underscoring the critical role of diverse datasets and transfer learning in promoting fairness.

Strong Points:

• The work does a relevant literature survey and provides a completed experiment to investigate the racial disparities in type 1 diabetes.

• It utilizes linear mixed-effects models to analyze model coefficients, providing deeper insights into the attributes influencing model performance.

• The application of the surveillance error grid effectively demonstrates the impact of prediction errors on diabetes patients, extending beyond a single RMSE metric.

• The paper is easy to read, and the model study design is well explained.

Weak Points & Suggestions:

• While the abstract highlights a slope difference between non-Hispanic White and Black groups, the discussion section lacks an explanation or description of this finding. The authors should elaborate on how this observation affects the fairness of glucose predictions.

• The study relies solely on the LSTM model for experimentation. Exploring additional machine learning models, such as traditional linear regression, multilayer perceptron, or advanced transformer-based models, could provide a broader understanding of potential racial disparities in healthcare AI applications.

• The study's limitations acknowledge the exclusive focus on non-Hispanic White and Black participants but overlook other important demographic variables, such as age strata. Including these factors could enhance the scope of the research.

• The lack of external dataset evaluation and the relatively small sample size may limit the generalizability of the findings. The authors are encouraged to incorporate external datasets with varied racial compositions to assess racial disparities across datasets.

6. PLOS authors have the option to publish the peer review history of their article (what does this mean? ). If published, this will include your full peer review and any attached files.

**Do you want your identity to be public for this peer review?** For information about this choice, including consent withdrawal, please see our Privacy Policy .

Reviewer #1: No

Reviewer #2: No

---

## [Decision Letter · Decision Letter 1]

6 June 2025

Racial disparities in continuous glucose monitoring-based 60-min glucose predictions among people with type 1 diabetes

PDIG-D-25-00023R1

Dear Dr. Hulman,

We are pleased to inform you that your manuscript 'Racial disparities in continuous glucose monitoring-based 60-min glucose predictions among people with type 1 diabetes' has been provisionally accepted for publication in PLOS Digital Health.

Best regards,

Gloria Hyunjung Kwak

Section Editor

PLOS Digital Health

**Additional Editor Comments (if provided):**

**Reviewer Comments (if any, and for reference):**

Reviewer's Responses to Questions

**Comments to the Author**

1. If the authors have adequately addressed your comments raised in a previous round of review and you feel that this manuscript is now acceptable for publication, you may indicate that here to bypass the “Comments to the Author” section, enter your conflict of interest statement in the “Confidential to Editor” section, and submit your "Accept" recommendation.

Reviewer #3: All comments have been addressed

Reviewer #4: All comments have been addressed

2. Does this manuscript meet PLOS Digital Health’s publication criteria ? Is the manuscript technically sound, and do the data support the conclusions? The manuscript must describe methodologically and ethically rigorous research with conclusions that are appropriately drawn based on the data presented.

Reviewer #3: Yes

Reviewer #4: Yes

3. Has the statistical analysis been performed appropriately and rigorously?

Reviewer #3: Yes

Reviewer #4: Yes

4. Have the authors made all data underlying the findings in their manuscript fully available (please refer to the Data Availability Statement at the start of the manuscript PDF file)?

Reviewer #3: Yes

Reviewer #4: Yes

5. Is the manuscript presented in an intelligible fashion and written in standard English?

Reviewer #3: Yes

Reviewer #4: Yes

6. Review Comments to the Author

Reviewer #3: I consider the paper a very relevant manuscript that addresses a critical issue in digital health. The investigation into racial disparities and the evaluation of transfer learning as a mitigation strategy are valid. The study design is well articulated, the analyses are appropriate, and your commitment to open science principles through public data and code sharing significantly strengthens the work. The ethical standards are followed. The findings demonstrate the importance of diverse training data and fairness-aware approaches in developing AI-driven healthcare tools, making this a valuable contribution to PLOS Digital Health and the broader efforts to ensure equitable digital healthcare solutions.

Reviewer #4: All comments have been addressed comprehensively.

7. PLOS authors have the option to publish the peer review history of their article (what does this mean? ). If published, this will include your full peer review and any attached files.

**Do you want your identity to be public for this peer review?** For information about this choice, including consent withdrawal, please see our Privacy Policy .

Reviewer #3: No

Reviewer #4: No
